# Prevalence and predictors of anemia among adults on highly active antiretroviral therapy in Northeast Ethiopia: A retrospective cohort study

Teklehaimanot Fentie Wendie[ID]*, Getnet Mengistu[ID]

Department of Pharmacy, Wollo University, Dessie, Ethiopia

* tklhmntfnt5@gmail.com, Teklehaimanot.fentie@wu.edu.et

## Abstract

### Background

Although antiretroviral therapy has significantly altered the natural history of human immunodeficiency virus infection and improved the quality of life of patients, there are conflicting reports regarding its impact on hematological outcomes. Thus, this study aimed at investigating the prevalence and predictors of anemia among adults on antiretroviral therapy in Northeast Ethiopia.

### Materials and methods

A retrospective cohort study was carried out among adults who began antiretroviral treatment between September 2005 and January 2019 at two governmental hospitals in Dessie town. Data were collected from patients' medical records using a pretested data extraction instrument. Anemia was the primary outcome variable of the study. It was defined based on WHO criteria after adjustment for altitude and smoking status of measured values. Data were entered and validated using EpiData Version 3.1 and then exported to SPSS Version 20.0 for analysis. Descriptive analysis was done for prevalence and binary logistic regression was carried out to assess whether covariates were associated with experiencing anemia. Statistical significance has been considered at p-value <0.05.

### Results

Medical records of 392 patients (mean age: 35.58 ± 9.46 years) were reviewed. Of the total 392 patients, 218 (55.6%) were females, 261 (66.6%) were categorized under WHO clinical stage III/IV and 134 (34.2%) had a baseline CD4 cell count of <100 cells/mm$^3$. The mean baseline CD4 cell count was 179 cells/mm$^3$ (range: 2 to 853 cells) and 230 (58.7%) of the participants were on zidovudine-based regimen. Anemia was diagnosed among 162 (41.3%) patients. After adjustment for other confounding factors, risk of anemia was significantly associated with low baseline CD4 cell count (AOR 1.80, 95% CI 1.05–3.06) and

**Data Availability Statement:** All relevant data are within the paper and its Supporting information files.

**Funding:** The author(s) received no specific funding for this work.

**Competing interests:** The authors have declared that no competing interests exist.

tenofovir based regimen (AOR 2.05, 95% CI 1.31–3.21). On the other hand, being educated was found to be protective (AOR 0.40, 95% CI 0.21–0.78).

## Conclusion

In this research, the prevalence of anemia was relatively high. Low baseline CD4 cell count and tenofovir based regimen were independent predictors of anemia; while being educated was protective. Treatment programs should focus on early diagnosis and treatment of HIV as well as routine screening and proper treatment of anemia.

## Introduction

The introduction of highly active antiretroviral therapy (HAART) in 1996 and the increasing access to these potent antiretroviral drugs has dramatically improved the rate of HIV disease progression and mortality [1]. However, adverse drug events like hematologic toxicities remain as common causes of morbidity and mortality in people receiving HAART [2, 3]. Anemia is the most commonly reported hematologic abnormality with substantial impact on quality of life and clinical outcomes like HIV disease progression [4–7].

Studies have consistently shown that the prevalence of anemia is high among people living with HIV, particularly among those with advanced disease stage [8, 9]. The prevalence of anemia during the HAART era greatly varies depending on region, clinical settings and threshold used to define anemia [5, 10]. The estimated prevalence of anemia among adults receiving HAART was 35% in Europe and North America [11]; 39.2% in China [12] and 66.7% in Nepal [13]. Other reports also revealed that the prevalence of anemia ranges from 23 to 50% globally and 24 to 58% in Africa [3, 14].

Review of the literature demonstrated high prevalence of anemia among HIV infected adults in Ethiopia with pooled prevalence of 31% [15]. Studies conducted at different times from different parts of the country reported that the prevalence of anemia ranges from 11.4 to 70.1% [16–21]. Accordingly, it was estimated and reported as 23 to 70.1% in Northwest Ethiopia [21]; 36.5 to 53.3% in Southern Ethiopia [17]; 11.4 to 34.6% in Addis Ababa [16, 18, 19]; and 41.2% in Eastern Ethiopia [20].

Anemia in HIV-infected individuals has been associated with reduced quality of life, rapid disease progression, and increased mortality [4, 5, 22–25]. Analysis of a multi-centered data in Europe demonstrated that a 1 g/dL reduction in Hb raised the hazard of death by 57%, after controlling viral load and CD4 lymphocyte count. Accordingly, the latest hemoglobin value was an independent prognostic factor for death [23]. Mortality was higher in the first year following initiation of HAART, with anemia being one of the associated factors suggesting the need for routine screening and treatment of anemia [23, 26]. Anemia has also been associated with rapid progression to AIDS and included in the disease-progression scoring system for patients on HAART [5, 25].

Although HAART has generally been accepted as the gold standard in HIV care [27], there are inconsistent reports regarding its impact on hematological complications [28]. Whereas a number of studies reported that highly active antiretroviral therapy could reduce anemia, others reported no improvement or even anemia as an adverse drug reaction of the antiretroviral therapy [27, 29–34].

Anemia in HIV infection has been reported to be multifactorial being the result of a number of interactive factors making it difficult to determine causality [35, 36]. Antiretroviral and

other drugs are also known to cause anemia [36–38]. Factors often reported to increase risk anemia include ART regimens containing zidovudine, more advanced disease stage or low CD4+ count, nutritional status or body mass index <18.5 kg/M$^2$, female sex, duration of HAART, presence of opportunistic infections, and low baseline hemoglobin level [8, 12, 13, 15–18, 21, 30, 39–42].

Previous studies investigated the prevalence and predictors of anemia among HIV infected patients in general. Moreover, its prevalence and predictors vary by setting and population among others. However, there is insufficient data about anemia after initiation of HAART among adults in northeast Ethiopia. Thus, this study aimed at filling this gap by investigating the prevalence and predictors of anemia among adults on HAART.

## Materials and methods

### Study design, area and period

A retrospective cohort study was carried out from September 2018 to January 2019 among adults on a standard antiretroviral treatment at ART clinics of Boru Meda and Dessie Referral Hospitals. These are the only governmental hospitals in Dessie town serving majority of clients from the Northeast part of the country. Dessie is located 400 km to the north of the capital Addis Ababa with an altitude between 2,470 and 2,550 meters above sea level. Besides testing and treatment of HIV/AIDS, the hospitals also provide general outpatient and inpatient services including medical, surgical, pediatric, psychiatric, Ophthalmic, Emergency, Gynecology and Obstetrics care.

### Study population and eligibility criteria

All HIV-positive patients aged 15+ who began ART between September 2005 and January 2018 and were treated for at least 3 months with pre- and post-ART hemoglobin values were eligible for study enrolment. On the other hand, patients who were already anemic at the time of ART initiation, transfer in/out cases, lost to follow up and those with incomplete medical records were excluded.

### Sample size determination and sampling procedure

Sample size was computed using single population proportion formula based on the following assumptions: true population proportion or estimated prevalence (P) = 0.5, absolute precision (d) = 0.05, and 95% confidence level. Because there was no previous similar study in the area, the true proportion of the population was assumed to be 50%. To account for incomplete medical records, a 2% contingency was added. Thus, the final sample size was 392. Participants were enrolled based on their refill appointment; thus, simple random sampling technique was used.

### Study variables

Anemia following at least three months of antiretroviral treatment, was the primary outcome variable of the study. Anemia was defined based on WHO criteria as a hemoglobin concentration of <12 g/dl for non-pregnant women and <13 g/dl for men after adjustment for altitude and smoking status of measured values [43]. Accordingly, 0.16 g/dL and 0.13 g/dL were subtracted from measured values for smokers and non-smokers, respectively. It was further classified as mild (11–11.9 g/dl for women and 11–12.9 g/dl for men), moderate (8–10.9 g/dl) and severe (<8 g/dl in both genders). Socio-demographic characteristics (age, gender, educational level), nutritional status, tuberculosis coinfection, WHO clinical stage, baseline CD4 cell

count, ART regimen, adherence, isoniazid and co-trimoxazole preventive therapies were the predictor variables.

## Data collection procedure

Socio-demographic characteristics, baseline and follow up clinical and laboratory data, as well as treatment outcomes were extracted from patients' medical records using a structured and pretested data abstraction format. Unique ART numbers were used for the identification of individual participants. The data abstraction tool has been adopted from ART registry and follow up forms of ART clinics and pretested on transferred-in cases before use. Patient ART registers, follow up medical records and the electronic dispensing database were used as data sources. Laboratory test reports were also used to collect hematologic, virologic and immunologic results. World health organization (WHO) cut off values of hemoglobin were taken after adjustment to altitude and smoking status to define and classify anemia. Study participants were retrospectively followed from the date of enrolment to treatment initiation.

## Data quality assurance

Pretest for the applicability of the data collection tool was done and training was given for the data collectors and supervisors before the commencement of the data collection procedure. Data were checked for completeness daily, coded, cleaned and verified before entry.

## Data processing and analysis

Data were entered using EpiData Version 3.1 and exported to SPSS Version 20.0 for analysis. Descriptive statistics were used for the analysis of baseline sociodemographic and clinical characteristics. Descriptive analysis was done for prevalence and binary logistic regression models were used to identify the predictors of anemia. The following variables were assessed as potential predictors: gender, age, residence, educational status, nutritional status, WHO clinical stage, tuberculosis coinfection, ART regimen, and baseline CD4 cell count. Odds ratios (OR) with the 95% confidence intervals were reported and statistical significance was considered at p-value $<0.05$.

## Ethical considerations

The study was carried out following approval by the institutional review committee of Wollo University and a subsequent permission from each hospital. Informed consent was waived by the Institutional Review Committee of the University and the hospitals due to the retrospective nature of the study, as all the data were collected from routine medical records. Confidentiality was guaranteed by omitting names or any personal identifiers. In addition, data were kept secured via out the research process to limit accessibility to a third party.

## Results

### Patient characteristics at enrollment

Medical records of 392 patients (mean age: 35.58 ± 9.46; range: 15–80) years were reviewed. Of the total 392 patients, 218 (55.6%) were females; 200 (51%) were married; 152 (38.8%) completed primary school; 214 (54.6%) were rural dwellers; 130 (33.2%) had a body mass index of $<18.5$ kg/m$^2$; 62 (15.8%) use social drugs like Khat, alcohol, or cigarette; 59 (15.1%) were tuberculosis coinfected; and 233 (59.4%) of them were on their initial ART regimen.

Clinically, 261 (66.6%) patients were categorized under WHO stage III/IV disease and 134 (34.2%) of them had a baseline CD4 cell count of $<100$ cells/mm$^3$ with mean cell count of 179

cells/mm$^3$ (range:2–853). Regarding standard prophylaxis against common opportunistic infections, 354 (90.3%) and 275 (70.2%) of the participants were treated with cotrimoxazole and isoniazid preventive therapies, respectively. The predominant ART regimen was Zidovudine + Lamivudine + Nevirapine and 230 (58.7%) of all participants were on Zidovudine based regimen (Table 1).

**Table 1. Baseline characteristics of patients on HAART in Northeast Ethiopia: 2005–2019.**

| | | Antiretroviral therapy Regimen | | | | Total |
|---|---|---|---|---|---|---|
| | | AZT/3TC/NVP | AZT/3TC/EFV | TDF/3TC/EFV | TDF/3TC/NVP | |
| Sex | Male | 52 | 44 | 64 | 14 | 174 (44.4%) |
| | Female | 108 | 26 | 57 | 27 | 218 (55.6%) |
| Age; mean: 35.58 | ≤35 years | 103 | 27 | 64 | 21 | 215 (54.8%) |
| | >35 years | 57 | 43 | 57 | 20 | 177 (45.2%) |
| Marital Status | Never Married | 30 | 17 | 23 | 11 | 81 (20.7%) |
| | Married | 75 | 35 | 69 | 21 | 200 (51%) |
| | Widowed | 18 | 9 | 13 | 3 | 43 (11%) |
| | Divorced | 37 | 9 | 16 | 6 | 68 (17.3%) |
| Education Level | Illiterate | 50 | 19 | 23 | 13 | 105 (26.8%) |
| | Primary school | 60 | 33 | 45 | 14 | 152 (38.8%) |
| | High school | 41 | 16 | 33 | 6 | 96 (24.5%) |
| | College/above | 9 | 2 | 20 | 8 | 39 (9.9%) |
| Residence | Rural | 98 | 37 | 58 | 21 | 214 (54.6%) |
| | Urban | 62 | 33 | 63 | 20 | 178 (45.4%) |
| Social drug use | None | 138 | 58 | 98 | 36 | 330 (84.2%) |
| | Yes | 22 | 12 | 23 | 5 | 62 (15.8%) |
| Tuberculosis test | Negative | 138 | 52 | 105 | 38 | 333 (84.9%) |
| | Positive | 22 | 18 | 16 | 3 | 59 (15.1%) |
| Experience to ART | Naïve* | 90 | 48 | 86 | 9 | 233 (59.4%) |
| | Non-naïve | 70 | 22 | 35 | 32 | 159 (40.6%) |
| WHO Clinical Stage | Stage I | 11 | 2 | 16 | 4 | 33 (8.4%) |
| | Stage II | 43 | 16 | 28 | 11 | 98 (25.0%) |
| | Stage III | 95 | 43 | 61 | 24 | 223 (56.9%) |
| | Stage IV | 11 | 9 | 16 | 2 | 38 (9.7%) |
| CD4 cell count in cells/ml$^3$; (Mean: 179) | >200 | 43 | 24 | 49 | 14 | 130 (33.2%) |
| | 100–200 | 56 | 18 | 38 | 16 | 128 (32.7%) |
| | <100 | 61 | 28 | 34 | 11 | 134 (34.2%) |
| Body Mass Index | <18.5 kg/m2 | 65 | 22 | 35 | 8 | 130 (33.2%) |
| | 18.5–25 kg/m2 | 83 | 48 | 79 | 30 | 240 (61.2%) |
| | >25 kg/m2 | 12 | 0 | 7 | 3 | 22 (5.6%) |
| Cotrimoxazole Prophylaxis | No | 8 | 10 | 13 | 7 | 38 (9.7%) |
| | Yes | 152 | 60 | 108 | 34 | 354 (90.3%) |
| Received Isoniazid | No | 43 | 26 | 36 | 12 | 117 (29.8%) |
| | Yes | 117 | 44 | 85 | 29 | 275 (70.2%) |
| **Total** | | **160 (40.8%)** | **70 (17.9%)** | **121 (30.9%)** | **41 (10.5%)** | **392** |

**Abbreviations**: NVP, nevirapine; EFV, efavirenz; AZT, zidovudine; TDF, tenofovir; 3TC, lamivudine;

* were on their initial ART regimen

**Table 2. Interim treatment outcome among adults on HAART in Northeast Ethiopia: 2005–2019.**

| | | Antiretroviral therapy Regimen | | | | Total |
|---|---|---|---|---|---|---|
| | | AZT/3TC/NVP | AZT/3TC/EFV | TDF/3TC/EFV | TDF/3TC/NVP | |
| Adherence | Good | 131 | 52 | 106 | 37 | 326 (83.2%) |
| | Fair/poor | 29 | 18 | 15 | 4 | 66 (16.8%) |
| Adverse Drug Reaction | No | 89 | 45 | 84 | 10 | 228 (58.2%) |
| | Yes* | 71 | 25 | 37 | 31 | 164 (41.8%) |
| Recent Retroviral Load | Not done | 19 | 6 | 7 | 2 | 34 (8.7%) |
| | Undetectable | 123 | 53 | 99 | 39 | 314 (83.1%) |
| | Detectable** | 18 | 11 | 15 | 0 | 44 (11.2%) |
| New onset Anemia | No | 98 | 49 | 59 | 24 | 230 (58.7%) |
| | Yes | 62 | 21 | 62 | 17 | 162 (41.3%) |
| Duration of follow up in months (mean:76) | <61 | 28 | 12 | 79 | 12 | 131 (33.4%) |
| | 61–90 | 64 | 12 | 30 | 25 | 131 (33.4%) |
| | >90 | 68 | 46 | 12 | 4 | 130 (33.2%) |
| Severity of anemia (n: 162) | Mild | 38 | 10 | 38 | 4 | 90 (55.6%) |
| | Moderate | 23 | 10 | 20 | 8 | 61 (37.6%) |
| | Severe | 1 | 1 | 4 | 5 | 11 (6.8%) |

\* Lipodystrophy, anemia, Central Nervous System disturbance, and/or rash,

\*\* >1000 copies/ml

## Interim treatment outcomes

Of the total 392 study participants, 164 (41.8%) had experienced at least one adverse drug reaction; 326 (83.2%) had good adherence to their prescribed medication; 314 (80.1%) achieved their virologic goal with undetectable viral load and 162 (41.3%) became anemic; of which 11 (6.8%) patients had experienced severe anemia requiring regimen change. The mean duration of follow-up on HAART was 76 ± 32.5 months (range: 7 to 150 months) (Table 2).

## Predictors of anemia

Bivariate analysis of sociodemographic and clinical variables revealed that low baseline CD4 cell count (crude odds ratio (COR) 1.66, 95% confidence interval (CI) 1.01–2.74) and TDF based ART regimen (COR 1.69, 95% CI 1.12–2.54) were associated with the risk of anemia. On the other hand, having education level of high school was found to be protective (COR 0.46, 95% CI 0.26–0.83). Similarly, multivariate logistic regression analysis confirmed that baseline CD4 cell count of 100–200 cells/mm$^3$ (adjusted odds ratio (AOR) 1.80, 95% CI 1.05–3.06 and TDF based ART regimen (AOR 2.05, 95% CI 1.31–3.21) were significant predictors of anemia; and being educated was found to be protective (AOR 0.40, 95% CI 0.21–0.78) (Table 3).

Patients with baseline CD4 cell count of 100–200 cells/mm$^3$ were about twice at risk of being anemic compared to those with CD4 cell count of >200 cells/mm$^3$ (AOR 1.80, 95% CI 1.05–3.06). Surprisingly, the risk of experiencing anemia was two times (AOR 2.05, 95% CI 1.31–3.21) among individuals on TDF based ART regimen than those who were on AZT based ART regimen. Patients who were educated up to high school were about 60% less at risk of facing anemia compared to those individuals who didn't attend school (AOR 0.40, 95% CI 0.21–0.78). However, no statistically significant association was found between anemia and the following covariates: sex, age, residency, body mass index, tuberculosis coinfection, WHO

**Table 3. Possible predictors of anemia among adults on HAART in Northeast Ethiopia: 2005–2019.**

| Covariates | Categories | Anemia frequency (%) | COR (95% C.I.) | AOR (95% C.I.) |
|---|---|---|---|---|
| Sex | Male | 66 (37.9%) | 1.00 | 1.00 |
| | Female | 96 (44.0%) | 1.29 (0.86, 1.93) | 1.22 (0.77, 1.92) |
| Age, mean: 35.58 | ≤35 years | 93 (43.3%) | 1.00 | 1.00 |
| | >35 years | 69 (39.0%) | 0.84 (0.56, 1.26) | 0.77 (0.49, 1.20) |
| Body mass index | 18.5–25 kg/m2 | 102 (42.5%) | 1.00 | 1.00 |
| | <18.5 kg/m2 | 52 (40.0%) | 0.90 (0.58, 1.39) | 0.88 (0.55, 1.40) |
| | >25 kg/m2 | 8 (36.4%) | 0.77 (0.31, 1.91) | 0.97 (0.36, 2.59) |
| Residence | Urban | 71 (39.9%) | 1.00 | 1.00 |
| | Rural | 91 (42.5%) | 1.12 (0.74, 1.67) | 0.96 (0.60, 1.53) |
| Education Level | Illiterate | 52 (49.5%) | 1.00 | 1.00 |
| | Primary school | 65 (42.8%) | 0.76 (0.46, 1.26) | 0.71 (0.41, 1.22) |
| | High school | 30 (31.2%) | 0.46 (0.26, 0.83) * | 0.40 (0.21, 0.78) * |
| | College & above | 15 (38.5%) | 0.64 (0.30, 1.35) | 0.46 (0.20, 1.07) |
| TB co-infection | No | 136 (40.8%) | 1.00 | 1.00 |
| | Yes | 26 (44.1%) | 1.14 (0.65, 1.99) | 1.01 (0.54, 1.90) |
| Experience to ART | Naïve | 92 (39.5%) | 1.00 | 1.00 |
| | Non-naïve | 70 (44.0%) | 1.21 (0.80, 1.81) | 1.08 (0.69, 1.69) |
| Baseline CD4 count | >200 cells/mm$^3$ | 46 (35.4%) | 1.00 | 1.00 |
| | 100–200 cells/mm$^3$ | 61 (47.7%) | 1.66 (1.01, 2.74) * | 1.80 (1.05, 3.06) * |
| | <100 cells/mm$^3$ | 55 (41.0%) | 1.27 (0.77, 2.09) | 1.43 (0.83, 2.47) |
| WHO clinical stage | Stage I or II | 52 (39.7%) | 1.00 | 1.00 |
| | Stage III or IV | 110 (42.1%) | 1.11 (0.72, 1.70) | 1.30 (0.80, 2.10) |
| ART Regimen | AZT Based | 83 (36.1%) | 1.00 | 1.00 |
| | TDF Based | 79 (48.8%) | 1.69 (1.12, 2.54) * | 2.05 (1.31, 3.21) * |
| Adherence | Good | 132 (40.5%) | 1.00 | 1.00 |
| | Fair/Poor | 30 (45.5%) | 1.23 (0.72, 2.09) | 1.24 (0.70, 2.19) |
| Isoniazid prophylaxis | No | 55 (47.0%) | 1.39 (0.90, 2.16) | 1.59 (0.98, 2.57) |
| | Yes | 107 (38.9%) | 1.00 | 1.00 |
| Co-trimoxazole prophylaxis | No | 15 (39.5%) | 1.00 | 1.00 |
| | Yes | 147 (41.5%) | 1.09 (0.55, 2.16) | 1.18 (0.58, 2.44) |

* Significantly associated with anemia

clinical stage, being naïve or non-naïve to ART, adherence, isoniazid and cotrimoxazole preventive therapies.

## Discussion

In the HAART era, anemia is still common and independently associated with increased mortality. Correction of anemia was associated with reversal of this increased risk. Identification of individuals at risk is crucial for early detection and appropriate management of antiretroviral toxicities like anemia. This study was aimed to investigate the prevalence and associated factors of anemia among adults on highly active antiretroviral therapy.

According to this study, the prevalence of anemia was 41.3% (95% CI: 36–46%) showing that anemia is still prevalent among patients receiving HAART. This finding was comparable with reports from Southern Ethiopia 36.5% [39], Uganda 47.8% [44] and China 39.2% [12]. Our finding was significantly lower than that of prior studies conducted in Uganda 67.4% [45] and Nepal 66.7% [13]. However; this result was higher than reports from different parts of

Ethiopia like Southern Ethiopia 34.8% [17], Addis Ababa 34.6% [16], Northwest Ethiopia 23–34% [21, 40, 46], and a national pooled prevalence of anemia 31.0% [15]. The observed discrepancy might be attributed to the differences in socioeconomic status, health literacy, study design, sample size, inclusion criteria, and most importantly the difference in cut-off values that might be used to define anemia.

Regarding the severity of anemia, majority of cases were mild (55.6%) with the remaining 37.6% to be moderate and only 6.8% of them were severe requiring regimen change. This finding was consistent with a report from rural China where 69.4%, 27.5% and 3.1% of all anemic patients were experiencing mild, moderate and severe anemia, respectively [12]. The prevalence of severe anemia was lower compared to that of previous studies conducted in Southern Ethiopia (21.8%) [17], Addis Ababa (14.5%) [16] and Nepal (17.8%) [13]. However, it was relatively higher than a report form Northwest Ethiopia 2.5% [46].

Multivariate logistic regression analysis shows that TDF based ART regimen and advanced disease (low baseline CD4 cell count) were independent predictors of anemia. Patients with baseline CD4 cells count of 100–200 cells/mm$^3$ were approximately twice more likely to be anemic compared to those individuals with a CD4 count of >200 cells/mm$^3$ (AOR 1.80, 95% CI 1.05–3.06). Similarly, studies conducted in Northwest Ethiopia [21, 40, 41, 46], Nepal [13], Uganda [44], and China [12] showed that low CD4 cell count was significantly associated with anemia. This was also supported by the result of a systematic review and meta-analysis done in Ethiopia [15]. The possible explanation for this association may be opportunistic infections, malnutrition, or HIV associated myelosuppression with advanced disease.

In this study, using Zidovudine (AZT) based ART regimen did not show significant association with anemia despite its frequent association with bone marrow suppression. Unusually, patients who took TDF-based ART regimens were twice more likely to be anemic than those who took AZT-based regimens (AOR 2.05, 95% CI 1.31–3.21). Although a similar finding was reported in Addis Ababa, Ethiopia [18], it is inconsistent with the finding of studies conducted elsewhere in Northwest Ethiopia [21, 40, 41]. Though it needs further investigation, it could be explained by anemia as a complication of TDF induced renal impairment.

This study also identified educational status as predictor of anemia among the study subjects. Patients who never attended school were more likely to suffer from anemia compared to those who completed high school. On the other hand, being educated up to high school was found to be protective (AOR 0.40, 95% CI 0.21–0.78) probably due to good health-seeking behavior or healthy diet patterns. The finding is supported by a report from Northwest Ethiopia [21].

However, no statistically significant association was observed between anemia and the following covariates: sex, age, residency, body mass index, tuberculosis coinfection, WHO clinical stage, being naïve or non-naïve to ART, adherence, as well as isoniazid and cotrimoxazole preventive therapies. These were inconsistent with the findings of previous studies done around the globe [12, 13, 15, 16, 21, 40, 41, 44–46]. Since the causes of anemia in HIV patients are multi-factorial, we recognize the inability of a single study to identify its specific causes exhaustively. Our study was not without limitation. The retrospective nature of this study might be a limitation as the accuracy of data depends on the completeness the records, and hence, information bias might have occurred because of underreporting/missing data elements.

## Conclusions

In this study, the prevalence of anemia was relatively high though previous studies reported a significant association between HAART and increment in hemoglobin. Low baseline CD4 cell count and TDF based ART regimen were found to be independent predictors of anemia; while

being educated was protective. ART programs should focus on early diagnosis and treatment of HIV as well as routine screening and proper treatment of anemia. Further studies are needed to identify prospectively patients at high risk and design interventions aimed to reduce this risk.

## Supporting information

**S1 File.**
(XLSX)

## Acknowledgments

We would like to express our sincere gratitude to the management officials and staff of the hospitals for their cooperation and data collectors for their commitment.

## Author Contributions

**Conceptualization:** Teklehaimanot Fentie Wendie.

**Data curation:** Teklehaimanot Fentie Wendie, Getnet Mengistu.

**Formal analysis:** Teklehaimanot Fentie Wendie.

**Investigation:** Teklehaimanot Fentie Wendie, Getnet Mengistu.

**Methodology:** Teklehaimanot Fentie Wendie, Getnet Mengistu.

**Software:** Teklehaimanot Fentie Wendie, Getnet Mengistu.

**Writing – original draft:** Teklehaimanot Fentie Wendie.

**Writing – review & editing:** Teklehaimanot Fentie Wendie, Getnet Mengistu.

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
