## [Decision Letter · Decision Letter 0]

10 Dec 2021

PONE-D-21-25369

Prevalence and predictors of anemia among adults on highly active antiretroviral therapy in Northeast Ethiopia: a retrospective cohort study

PLOS ONE

Dear Dr. Fentie,

Thank you for submitting your manuscript to PLOS ONE. After careful consideration, we feel that it has merit but does not fully meet PLOS ONE’s publication criteria as it currently stands. Therefore, we invite you to submit a revised version of the manuscript that addresses the points raised during the review process.

We look forward to receiving your revised manuscript.

Kind regards,

Professor Kwasi Torpey, MD PhD MPH

Academic Editor

PLOS ONE

Journal Requirements:

- https://www.dovepress.com/prevalence-and-predictors-of-virological-failure-among-adults-living-w-peer-reviewed-fulltext-article-HIV

In your revision ensure you cite all your sources (including your own works), and quote or rephrase any duplicated text outside the methods section. Further consideration is dependent on these concerns being addressed.

Reviewers' comments:

Reviewer's Responses to Questions

**Comments to the Author**

1. Is the manuscript technically sound, and do the data support the conclusions?

Reviewer #1: Yes

Reviewer #2: Yes

2. Has the statistical analysis been performed appropriately and rigorously? 

Reviewer #1: Yes

Reviewer #2: Yes

3. Have the authors made all data underlying the findings in their manuscript fully available?

Reviewer #1: Yes

Reviewer #2: Yes

4. Is the manuscript presented in an intelligible fashion and written in standard English?

Reviewer #1: Yes

Reviewer #2: Yes

5. Review Comments to the Author

Reviewer #1: The duration of the study is between September 2005 to January 2018 in the abstract but its September 2005 to January 2019 in the methodology. Which is which?

It’s quite difficult to understand when the hemoglobin measurement was taken. The paper does not clearly state the timelines for hemoglobin measurements.

The stratification of the patients is not clear. The authors need to clearly define how their subjects were selected; what parameter was measured at what time. Was there baseline measurements and subsequent measurements of hemoglobin? Were all the subjects on ART at certain point? What was the point of including the naïve subjects when the work was clearly looking at effect of HAART on hemoglobin levels?

Anemia is multifactorial in nature. How were the authors able to control for other confounders which could affect the level of hemoglobin?

Why was the TDF based ART regimen associated with anemia in the study? No reason was assigned to this in the discussion.

Why was there no bivariate analysis between duration of HAART and the risk of developing anemia?

Reviewer #2: Reviewer Report to the author (s)

This MS might give important information about Prevalence and predictors of anemia among adults on highly active antiretroviral therapy for clinicians and for the scientific community. But the authors need to address the following points before publication:

The authors name in the cover page can be rewrite as “Teklehaimanot Fentie Wendie1*, Kassahun Bogale1, Getnet Mengistu1”

The corresponding author Email address in the cover page can be modifies as :

E-mail: tklhmntfnt5@gmail.com(TF) (Teklehaimanot.fentie@wu.edu.et)

In the abstract, method subsection, the authors need to indicate the specific study sites at the end of the study period: A retrospective cohort study was carried out among adults who began highly active antiretroviral treatment between September 2005 and January 2018 in ----------.

From the abstract, the authors need to avoid the use of abbreviations (Eg. HARRT, ART, TDF, etc) before indicating their full names.

In this MS, there is unnecessary excessive use of references throughout the document after single sentences. The authors need to be specific and better to use only recent and relevant one or two references after a sentence of paragraphs.

The introduction part needs some rearrangements for simplicity for readers. The authors first indicate the global burden of anemia on HIV positive individuals, then in African and Ethiopia and the study area particularly.

The title or the subtitles may not be necessarily indicated on separate pages. For example, the methodology part need to be indicated immediately after introduction part in the same page.

The title “Method and participants” can be modified as “Methodology” or “Methods and Materials”. Based on this modification, the subtitle “methods” in the abstract section can be modified.

In your methodology part, the study area is not well elaborated. In addition, the authors didn’t clearly indicate the reason why only two government hospitals were selected among the others.

Since your title focuses to see the condition of anemia among HIV positive individuals who are on HAART, it is not clear the reason why you include study participants started from the age of 15 years. In Ethiopian context, the word “adult” refers individuals’ grater or equal to 18 years. Please reconsider it.

Based on your sample size assumptions, the calculated sample size will be around 383(4). But as indicated in your result, the final sample size was 392. It needs justification.

The hemoglobin level was evaluated after adjusting the attitude and smoking status. But it is not clearly indicated in the methods part. It seems theoretical assumption.

It is better to indicate where the pretest of the data extraction tool was tested. It is not clearly indicated.

In your ‘ethical consideration” subtitle, the phrase “….in compliance with the Helsinki Declaration on the ethical principles of medical research involving human subject…” is not important. Better to be deleted.

Since your title is among adults on highly active antiretroviral therapy, how 233 (59.4%) of them were naïve to ART in your result? It is very confusing.

In result at the time of enrollment, what was the prevalence of anemia? It is not indicated. But is very important for comparison to see the prevalence of anemia before and after the initiation of ART.

It is better to insert the respective tables immediately after the text it refers.

Indicating the number of study participants (N=392) at the end of the titles of the table is not convincing for the reader; it is better to be deleted.

Since the authors already indicated the meaning of “COR” and “AOR” in “Predictors of anemia” subtitle, no need of indicating them at the end of table 3 as key word. This concept can also hold true for the other abbreviations indicated in the respective tables.

In your discussion part, second paragraph, the phrase “….defined as Hgb concentration <13 g/dl for males and <12 g/dl for females after adjustment for altitude and smoking status of measured values…” is not important. It is almost reputation. Better to be deleted.

When you compare results in discussion, you need to have 95% CI in your result. For example the prevalence of anemia in your result is 41.3%. But you didn’t indicate its 95%CI. Hence it is very difficult to say comparable, lower or higher than other reports. This can be true for other variables that you compare.

From your conclusion, avoid unnecessary explanations. For example avoid “(100 - 200 cells/mm3)”

6. PLOS authors have the option to publish the peer review history of their article (what does this mean?). If published, this will include your full peer review and any attached files.

Reviewer #1: **Yes: **Christian Obirikorang

Reviewer #2: **Yes: **Dr. Yitayih Wondimeneh

---

## [Author Response · Author response to Decision Letter 0]

23 Jan 2022

Author's Response to reviewers & editorial corrections

Title: "Prevalence and predictors of anemia among adults on highly active antiretroviral therapy in Northeast Ethiopia: a retrospective cohort study" (Manuscript ID: PONE-D-21-25369)

Authors:

Teklehaimanot Fentie Wendie (tklhmntfnt5@gmail.com)

Getnet Mengistu (mgetnet12@gmail.com)

Thank you for consideration of our manuscript for publication in your journal and our sincere gratitude goes to your esteemed reviewers. We have revised the above manuscript as per the comments given by the Reviewers and the requested Editorial corrections.

Journal Requirements:

Please ensure that your manuscript meets PLOS ONE's style requirements, including those for file naming. 

o We have prepared (edited) the manuscript using PLOS ONE style templates.

In your Data Availability statement, you have not specified where the minimal data set underlying the results described in your manuscript can be found. Upon re-submitting your revised manuscript, please upload your study’s minimal underlying data set as either Supporting Information files or to a stable, public repository and include the relevant URLs, DOIs, or accession numbers within your revised cover letter. 

o The raw data of the study was uploaded as Supporting Information file 1. This is stated in the data availability statement of the manuscript.

Regarding the occurrence of overlapping text with a previous publication and the need for language proof reading,

o A major revision was made as needed.

Reviewers' comments:

Reviewer #1: 

The duration of the study is between September 2005 to January 2018 in the abstract but its September 2005 to January 2019 in the methodology. Which is which?

o Sorry for the editorial error! It is January 2019.

It’s quite difficult to understand when the hemoglobin measurement was taken. The paper does not clearly state the timelines for hemoglobin measurements.

o Baseline (at time of ART initiation) and recent follow up hemoglobin measurements were taken and adjusted for altitude and smoking status based on WHO recommendations to determine whether the patient became anemic or not. 

The stratification of the patients is not clear. The authors need to clearly define how their subjects were selected; what parameter was measured at what time. 

o Was there baseline measurements and subsequent measurements of hemoglobin? Yes 

o Were all the subjects on ART at certain point? Yes 

o What was the point of including the naïve subjects when the work was clearly looking at effect of HAART on hemoglobin levels? The expression is somewhat confusing. Actually, the intention was to mean “were not on any ART regimen other than the current regimen” not to mean that “they are not on ART”. This is restated in the main text as “were on their initial ART regimen”.

Anemia is multifactorial in nature. How were the authors able to control for other confounders which could affect the level of hemoglobin?

o As much as possible, we have tried to incorporate all potential predictors of anemia like gender, age, residence, educational status, nutritional status, WHO clinical stage, tuberculosis coinfection, ART regimen, and baseline CD4 cell count and binary logistic regression models were used to control possible confounders and identify the predictors of anemia.

Why was the TDF based ART regimen associated with anemia in the study? No reason was assigned to this in the discussion.

o This finding was also a surprise to the authors. Though it needs further investigation, it could be explained by anemia as a complication of TDF induced renal impairment. 

Why was there no bivariate analysis between duration of HAART and the risk of developing anemia?

o Duration of HAART was assessed, but only those covariates with P-value of less than or equal to 0.25 were included in the multivariate analysis.

Reviewer #2: Reviewer Report to the author (s)

The authors name in the cover page can be rewrite as “Teklehaimanot Fentie Wendie1*, Kassahun Bogale1, Getnet Mengistu1”

o Rewritten as suggested.

The corresponding author Email address in the cover page can be modifies as :

E-mail: tklhmntfnt5@gmail.com (TF) (Teklehaimanot.fentie@wu.edu.et)

o Comment accepted

In the abstract, method subsection, the authors need to indicate the specific study sites at the end of the study period: A retrospective cohort study was carried out among adults who began highly active antiretroviral treatment between September 2005 and January 2018 in ----------.

o Inserted accordingly (….at two governmental hospitals in Dessie town)

From the abstract, the authors need to avoid the use of abbreviations (Eg. HARRT, ART, TDF, etc) before indicating their full names.

o Edited 

In this MS, there is unnecessary excessive use of references throughout the document after single sentences. The authors need to be specific and better to use only recent and relevant one or two references after a sentence of paragraphs.

o Thank you very much for your valuable comments; we have tried to limit the number of references by retaining only the pertinent ones. As can be seen in the main text, the number of references is reduced from 72 to 46.

The introduction part needs some rearrangements for simplicity for readers. The authors first indicate the global burden of anemia on HIV positive individuals, then in African and Ethiopia and the study area particularly.

o The introduction is stratified by prevalence, impact and predictors of anemia and paragraphs are structured from global to local.

The title or the subtitles may not be necessarily indicated on separate pages. For example, the methodology part need to be indicated immediately after introduction part in the same page.

o Comment accepted

The title “Method and participants” can be modified as “Methodology” or “Methods and Materials”. Based on this modification, the subtitle “methods” in the abstract section can be modified.

o Modified as “Materials and methods”.

In your methodology part, the study area is not well elaborated. In addition, the authors didn’t clearly indicate the reason why only two government hospitals were selected among the others.

o We have revised this section. The reason why only two government hospitals were selected among the others is that HIV testing and treatment is mainly provided in government hospitals and these are the only government hospitals serving majority of the clients from the Northeast part of the country.

Since your title focuses to see the condition of anemia among HIV positive individuals who are on HAART, it is not clear the reason why you include study participants started from the age of 15 years. In Ethiopian context, the word “adult” refers individuals’ grater or equal to 18 years. Please reconsider it.

o Individuals grater or equal to 18 years are considered adults for “legal issues” but for “health perspectives” especially in ART programs, individuals greater or equal to 15 years are treated as adults per WHO and national ART guidelines. 

Based on your sample size assumptions, the calculated sample size will be around 383(4). But as indicated in your result, the final sample size was 392. It needs justification.

o Right; the calculated sample size is 384. But, the final sample size was 392 because a 2% contingency was added to account for incomplete medical records.

The hemoglobin level was evaluated after adjusting the attitude and smoking status. But it is not clearly indicated in the methods part. It seems theoretical assumption.

o Measured haemoglobin concentrations were adjusted for altitude and smoking status based on WHO recommendations (1). Accordingly, 0.16 g/dL and 0.13 g/dL were subtracted from measured values for smokers and non-smokers, respectively. This is incorporated in the method. Remember that the study area has an altitude between 2,470 and 2,550 meters above sea level.

It is better to indicate where the pretest of the data extraction tool was tested. It is not clearly indicated.

o The data collection tool was pretested on transferred-in cases as already stated in the data collection procedure.

In your ‘ethical consideration” subtitle, the phrase “….in compliance with the Helsinki Declaration on the ethical principles of medical research involving human subject…” is not important. Better to be deleted.

o Comment accepted

Since your title is among adults on highly active antiretroviral therapy, how 233 (59.4%) of them were naïve to ART in your result? It is very confusing.

o “naïve to ART” is to mean “were not on any ART regimen other than the current regimen” not to mean that “they are not on ART”. This is restated as “were on their initial ART regimen”.

In result at the time of enrolment, what was the prevalence of anemia? It is not indicated. But is very important for comparison to see the prevalence of anemia before and after the initiation of ART.

o The intention of the study was to investigate the prevalence of anemia after the initiation of ART. So, Patients who were already anemic at the time of ART initiation were excluded as clearly stated in the eligibility criteria.

It is better to insert the respective tables immediately after the text it refers.

o Rearranged

Indicating the number of study participants (N=392) at the end of the titles of the table is not convincing for the reader; it is better to be deleted.

o Deleted 

Since the authors already indicated the meaning of “COR” and “AOR” in “Predictors of anemia” subtitle, no need of indicating them at the end of table 3 as key word. This concept can also hold true for the other abbreviations indicated in the respective tables.

o Comment accepted and acted up on it.

In your discussion part, second paragraph, the phrase “….defined as Hgb concentration <13 g/dl for males and <12 g/dl for females after adjustment for altitude and smoking status of measured values…” is not important. It is almost reputation. Better to be deleted.

o Thank you very much for your critical comments. Deleted. 

When you compare results in discussion, you need to have 95% CI in your result. For example the prevalence of anemia in your result is 41.3%. But you didn’t indicate its 95%CI. Hence it is very difficult to say comparable, lower or higher than other reports. This can be true for other variables that you compare.

o We have incorporated respective 95% confidence intervals. 

From your conclusion, avoid unnecessary explanations. For example avoid “(100 - 200 cells/mm3)”

o Deleted as commented.

1. Organization WH. Haemoglobin concentrations for the diagnosis of anaemia and assessment of severity. World Health Organization, 2011.

---

## [Decision Letter · Decision Letter 1]

1 Mar 2022

Prevalence and predictors of anemia among adults on highly active antiretroviral therapy in Northeast Ethiopia: a retrospective cohort study

PONE-D-21-25369R1

Dear Mr Fentie,

We’re pleased to inform you that your manuscript has been judged scientifically suitable for publication and will be formally accepted for publication once it meets all outstanding technical requirements.

Kind regards,

Professor Kwasi Torpey, MD PhD MPH

Academic Editor

PLOS ONE

Additional Editor Comments (optional):

Comments adequately addressed

Reviewers' comments:

Reviewer's Responses to Questions

**Comments to the Author**

1. If the authors have adequately addressed your comments raised in a previous round of review and you feel that this manuscript is now acceptable for publication, you may indicate that here to bypass the “Comments to the Author” section, enter your conflict of interest statement in the “Confidential to Editor” section, and submit your "Accept" recommendation.

Reviewer #1: (No Response)

2. Is the manuscript technically sound, and do the data support the conclusions?

Reviewer #1: Yes

3. Has the statistical analysis been performed appropriately and rigorously? 

Reviewer #1: Yes

4. Have the authors made all data underlying the findings in their manuscript fully available?

Reviewer #1: Yes

5. Is the manuscript presented in an intelligible fashion and written in standard English?

Reviewer #1: Yes

6. Review Comments to the Author

Reviewer #1: (No Response)

7. PLOS authors have the option to publish the peer review history of their article (what does this mean?). If published, this will include your full peer review and any attached files.

Reviewer #1: **Yes: **Christian Obirikorang

---

## [Editor Report · Acceptance letter]

18 Mar 2022

PONE-D-21-25369R1 

*Prevalence and predictors of anemia among adults on highly active antiretroviral therapy in Northeast Ethiopia: a retrospective cohort study*

Dear Dr. Fentie Wendie:

I'm pleased to inform you that your manuscript has been deemed suitable for publication in PLOS ONE. Congratulations! Your manuscript is now with our production department. 

Kind regards, 

on behalf of

Professor Kwasi Torpey 

Academic Editor

PLOS ONE